# Cerebroventricular Injection of Pgk1 Attenuates MPTP-Induced Neuronal Toxicity in Dopaminergic Cells in Zebrafish Brain in a Glycolysis-Independent Manner

**DOI:** 10.3390/ijms23084150

**Published:** 2022-04-08

**Authors:** Cheng-Yung Lin, Hsiang-Chien Tseng, Yu-Rong Chu, Chia-Lun Wu, Po-Hsiang Zhang, Huai-Jen Tsai

**Affiliations:** 1Institute of Biomedical Sciences, Mackay Medical College, New Taipei City 25245, Taiwan; cylin@mmc.edu.tw (C.-Y.L.); azx88561@gmail.com (Y.-R.C.); allen.wuu@gmail.com (C.-L.W.); wzmf173814@gmail.com (P.-H.Z.); 2Department of Anesthesiology, Shin Kong Wu Ho-Su Memorial Hospital, Taipei 11101, Taiwan; rockertw@gmail.com; 3School of Medicine, Fu Jen Catholic University, New Taipei City 20206, Taiwan; 4Department of Life Science, Fu Jen Catholic University, New Taipei City 20206, Taiwan

**Keywords:** dopamine neuron, CNS, zebrafish, Pgk1

## Abstract

Parkinson’s disease (PD) is characterized by the degeneration of dopaminergic neurons. While extracellular Pgk1 (ePgk1) is reported to promote neurite outgrowth, it remains unclear if it can affect the survival of dopaminergic cells. To address this, we employed cerebroventricular microinjection (CVMI) to deliver Pgk1 into the brain of larvae and adult zebrafish treated with methyl-4-phenyl-1,2,3,6-tetrahydropyridine (MPTP) as a PD-like model. The number of dopamine-producing cells in ventral diencephalon clusters of Pgk1-injected, MPTP-treated embryos increased over that of MPTP-treated embryos. Swimming distances of Pgk1-injected, MPTP-treated larvae and adult zebrafish were much longer compared to MPTP-treated samples. The effect of injected Pgk1 on both dopamine-producing cells and locomotion was time- and dose-dependent. Indeed, injected Pgk1 could be detected, located on dopamine neurons. When the glycolytic mutant Pgk1, Pgk1-T378P, was injected into the brain of MPTP-treated zebrafish groups, the protective ability of dopaminergic neurons did not differ from that of normal Pgk1. Therefore, ePgk1 is functionally independent from intracellular Pgk1 serving as an energy supplier. Furthermore, when Pgk1 was added to the culture medium for culturing dopamine-like SH-SY5Y cells, it could reduce the ROS pathway and apoptosis caused by the neurotoxin MPP^+^. These results show that ePgk1 benefits the survival of dopamine-producing cells and decreases neurotoxin damage.

## 1. Introduction

Parkinson’s disease (PD) is one of the most common neurodegenerative diseases. The main cause of PD is the formation of the Lewy body composed of aggregates of α-synuclein, resulting in the degeneration and death of dopaminergic neurons in substantia nigra pars compacta (SNpc). PD patients are diagnosed with clinical motor manifestations, including tremor, rigidity, and bradykinesia, as well as nonmotor features, including dysautonomia, cognitive dysfunction, dementia, sleep disturbances, and change of mood or behavior [1,2]. The pathological mechanisms and symptoms of PD include mitochondrial dysfunction and upregulated oxidative stress, eliciting the imbalance, or overexpression, of reactive oxygen species (ROS) [3], which, in turn, results in apoptosis [4]. Although many factors that include aging, gene mutation, and environmental toxins can cause PD, the complicated pathogenic mechanism of PD has never been fully elucidated [5]. Nowadays, medicinal, genetic, cellular, and surgical therapies have been utilized to alleviate the course of PD and the consequences of side effects, but the core pathology remains [6]. Therefore, further studies are necessary to unravel the pathogenic and molecular mechanism of PD.

The 1-methyl-4-phenyl-1,2,3,6-tetrahydropyridine (MPTP) is widely used to create a mimic PD animal model. Thus, PD-like symptoms can be observed in MPTP-treated animals, such as flies, worms, fish, mammals, primates, and even humans [7]. MPTP can pass through the blood–brain barrier (BBB), followed by oxidation and conversion to neurotoxic MPP^+^ via monoamine oxidase B (MAO-B) in glial neurons, the only cells containing this enzyme. Additionally, the structure of MPP^+^ is similar to dopamine, which can enter dopaminergic neurons through the dopamine transporter (DAT). After MPP^+^ enters neurons, it can block the mitochondrial electron transport chain, which, in turn, leads to cell damage and death caused by oxidative stress [8].

Compared with the invertebrates, zebrafish, the lower vertebrates, are more similar to humans in terms of genetic and protein functions. For example, the zebrafish genome has 71.4% homology with the human, and 82% of human disease genes have been defined in the zebrafish genome [9]. Based on anatomy and molecular markers, the dopamine neurons in the zebrafish ventral diencephalon (vDC), equivalent to human SN, send projection to the subpallium of the ventral telencephalon which is proposed to be homologous to the human striatum [10,11]. Having these important characteristics, the zebrafish becomes a good alternative animal model, serving to study human PD pathophysiology [12]. Additionally, zebrafish can be used as an animal model for studying the function of dopamine neurons at the early stages of embryos. The dopamine cells of zebrafish were detected in the vDC at 18 h post-fertilization (hpf) and most dopamine cells were developed at 96 hpf [13,14]. At 30 hpf, multiple dopamine receptors (D1-D4) were detected in the brain, indicating the dopamine signals were transmitted [15,16,17]. The number of dopamine neurons in vDC at 4 and 6 days post-fertilization (dpf) could also be defined using tyrosine hydroxylase (TH) staining [11,18]. Moreover, the zebrafish transgenic line *Tg(dat:EGFP),* the green fluorescence reporter driven by the dopamine active transporter (Dat) gene promoter [19], is available to make non-invasive imaging techniques to study the neuronal integrity possible, due to its transparent embryos.

For the PD study, zebrafish possess high homology and similar functions of genes corresponding with human genetic mutants which cause Parkinson related diseases [20], such as SNCA (zebrafish as SNCB, SNCG1, and SNCG2) [21], Parkin [22], PINK1 [23], DJ-1 [24], and LRRK2 [25]. Additionally, the zebrafish could be induced to suffer from a PD-like disease using environmental neurotoxins, such as MPTP, 6-OHDA, and Rotenone. This PD-like zebrafish model provides valuable information on the pathophysiological mechanism of PD [12,20,26]. For example, the number of dopamine neurons and swimming distance of 3-dpf zebrafish embryos were reduced in the PD-like zebrafish model [27]. These symptoms were also found in adult zebrafish where the intraperitoneal injection and cerebroventricular injection of MPTP were employed [28,29]. Moreover, not only was a systematic assessment method used to evaluate the movement disorders of zebrafish developed [30], but a large-scale drug screening platform for zebrafish was also generated [31,32].

Phosphoglycerate kinase 1 (Pgk1) is an enzyme that plays an important role in the glycolytic pathway in organisms. It can catalyze 1,3-bisphosphoglycerate into 3-phosphoglycerate and produce one unit of ATP [33]. Therefore, defective or mutated Pgk1 may result in Pgk1 deficiency disease, which is representative of an X-linked genetic disorder [34]. Most Pgk1 deficiency patients are frequently diagnosed with symptoms of early-onset PD, accompanied by the dysfunction of dopaminergic neurons [35,36,37]. Morales-Briceño et al. pointed out a potential causal relationship between Pgk1 deficient patients diagnosed with PD symptoms and impaired energy metabolisms [35]. However, Sakaue et al. reported that heterozygous female carriers of the Pgk1 mutation could present PD symptoms, even with the kinase activity of Pgk1 close to normal [36]. Thus, the detailed pathology and molecular mechanism between early-onset PD and Pgk1 deficiency require elucidation. 

Although the intracellular function involved in glycolysis of Pgk1 is well known, Lin et al. reported that Pgk1 can be secreted from the cells and perform a noncanonical function [38]. More specifically, we demonstrated that extracellular Pgk1 not only improves neurite outgrowth of NSC34 cells in vitro but also rescues the degeneration of motor neurons in vivo to maintain NMJ structure and mitigate progressive loss of motor ability, both in ALS mice and zebrafish [38]. Notably, Lin et al. [38] revealed that ePgk1 triggers the reduced phosphorylation of Cofilin at Ser3 (p-Cofilin-S3), a hallmark of growth cone collapse in neuronal cells, through decreasing the signaling pathway of Rac-GTP/p-Pak1-T423/p-MK2-T334/p-Limk1-S323/p-Cofilin-S3, which, in turn, enhances neurite outgrowth of motor neurons in a manner functionally independent from its intracellular, canonical role as a supplier of energy. This evidence suggests that the level of p-Cofilin-S3 is a biomarker determinative of neuronal cell development. Interestingly, Tseng et al. [39] reported that the level of p-Cofilin was increased in MPP^+^-treated primary mesencephalic cells, suggesting that it serves as a candidate biomarker of MPP^+^-induced neurite length reduction. Therefore, it would be worthwhile to investigate whether the addition of ePgk1 enables the reduced expression of p-Cofilin in MPP^+^/MPTP-treated cells, resulting in the alleviation neuronal damage. In this study, we demonstrated that the extracellular administration of Pgk1 could also serve as a neuron-protective substance for neurotoxin-treated dopamine neurons in the brain. It would be very interesting to conduct further investigation of this issue since a conclusive finding would be a significant step forward in understanding whether ePgk1 could also protect dopaminergic neurons from the degeneration and cell death that occurs in the brain of PD patients.

## 2. Results

### 2.1. Effect of MPTP on the Survival Rate and Swimming Ability of Zebrafish Larvae

To test the addition of ePgk1 as a potential therapy for PD, we used a mimic PD model of zebrafish established by treating embryos with MPTP. First, we studied the effect of different concentrations of MPTP, ranging from 30 to 180 μM, on the survival rate and swimming ability of zebrafish larvae (Figure 1). After the embryos were treated with MPTP from 2 to 6 dpf, the survival rate of the 180 μM-MPTP-treated larvae at 6 dpf was decreased to 55.3% (*n* = 25), while that of the 30-, 45-, and 90 μM-MPTP-treated larvae stayed at 100% (Figure 1A). Meanwhile, compared to the total swimming distance (4.46 cm) of WT larvae at 6 dpf, we found that higher concentrations of MPTP correlated with decreased swimming ability. Specifically, the total swimming distance of larvae treated with 90 μM of MPTP was 0.80 cm, suggesting that the swimming ability of larvae was severely affected by 90 μM of MPTP, although no death of larvae was observed. On the other hand, the swimming ability of larvae treated with 30 μM of MPTP exhibited little effect. Therefore, the 45 μM MPTP treatment was chosen for subsequent experiments since the swimming ability of larvae at this concentration was highly, but not drastically, affected (Figure 1B).

### 2.2. Cerebroventricular Microinjection (CAMI) of Pgk1 Prevented Dopaminergic Cells’ Death in the Ventral Diencephalon of Zebrafish Embryos Treated with MPTP

Under in vivo systems, the blood–brain barrier (BBB) poses numerous obstacles to the study of drug/protein/peptide efficacy against such CNS diseases as brain tumor, Parkinson’s disease, and Alzheimer’s disease in the brain of mammals [40]. The structure and function of the zebrafish BBB are reported to be similar to those of mammals [41]. Moreover, Quiñonez-Silvero et al. [42] reported that the zebrafish BBB begins to develop hindbrain vascularization in embryos at 28–32 h post-fertilization (hpf). After 72 hpf, the permeability of the vessels is quite limited, owing to the BBB. For example, DAPI (350 Da), Rhodamine-Dextran (10 kDa), and horseradish peroxidase (HRP, 44 kDa) are impermeable to cerebral microvessels in zebrafish embryos [43]. Since the development and function of zebrafish BBBs are well documented, we had to employ CVMI on zebrafish embryos and adults to circumvent such problems as those noted above. Nevertheless, before we could determine if exogenous Pgk1 was present in the brain chamber after CAMI, it was necessary to briefly perform the movement of protein across the BBB using the red fluorescence protein (DsRed), which served as a test material in this study. After the injection of DsRed into the brain chamber, the pattern of its fluorescence signal was individually observed in 2- and 4-dpf embryos, as well as in six-month-old adult zebrafish samples (Figure 2). The distribution pattern of DsRed in the 2-dpf embryos was apparent at CNS and the body close to the embryo head (Figure 2A,B). However, DsRed was distributed throughout the whole embryo for one hour after injection. In contrast, it was only dispersed in the brain chamber in 4-dpf embryos (Figure 2C–F). Meanwhile, in a parallel experiment, we created a small incision above the anterior portion of the optic tectum of a six-month-old adult zebrafish without damaging the brain (Figure 2G). After injecting DsRed into the cerebroventricular fluid surrounding the brain (Figure 2H,I), the red fluorescent signal was also only distributed evenly in the brain chamber for 30 min after injection (Figure 2J). Thus, since DsRed appeared at CNS and whole body of 2-dpf embryos, these results could only be explained by the incomplete formation of the BBB.

To study the effect of ePgk1 on the number of dopaminergic neurons in the ventral diencephalon (vDC) region of MPTP-treated zebrafish embryos, we counted the number of dopaminergic neurons, using immunoreaction with tyrosine hydroxylase (TH)-specific antibody labeled by green fluorescence signal (Figure 3A) at 6 dpf. When embryos were treated with MPTP from 2 to 6 dpf, Pgk1 was individually injected into the brain of embryos at 2, 4, and 2 plus 4 (2 + 4) dpf (Figure 3B). The average number of dopaminergic neurons in the ventral diencephalon (vDC) region of 6-dpf zebrafish larvae in the negative control (embryos untreated with MPTP but injected with DsRed at 2 dpf) displaying the green fluorescence signal was 74.5 ± 2.64 (*n* = 9) (Figure 3C,H), which was significantly higher than 41.6 ± 2.33 (*n* = 5), which was obtained from the mock control (embryos treated with MPTP from 2 to 6 dpf and injected with DsRed at 2 dpf) (Figure 3D,H). In the parallel experiment, the average number of dopaminergic neurons in vDC of MPTP-treated zebrafish embryos from 2 to 6 dpf, but injected with Pgk1 at 2, 4, and 2 plus 4 (2 + 4) dpf, was 43.25 ± 2.48 (*n* = 6), 48 ± 0.89 (*n* = 5) and 55.4 ± 1.36 (*n* = 5), respectively (Figure 3E–H).

Furthermore, we detected the expression of TH and GFP proteins in the head of larvae from the transgenic line *Tg(dat:EGFP)*, in which GFP is specifically expressed in dopaminergic neurons treated in the same experimental conditions as those described above. In addition, the Pgk1 glycolysis kinase-deficient mutant (Pgk1-T378P) was also employed to the (2 + 4) dpf group. Results demonstrated that the expression level TH protein in the DsRed-injected MPTP mock group was substantially decreased by 53% (*n* = 5) compared to that of the untreated negative control, which was set as 1 (Figure 3I,J). However, the expression level of TH protein was increased by 7.8, 12.1, 27.0, and 17.0% (*n* = 5) in the MPTP-treated embryos injected with Pgk1 at 2, 4, and (2 + 4) dpf, and with the Pgk1 mutant (Pgk1-T378P) at (2 + 4) dpf, respectively, compared to that of the DsRed-injected MPTP-mock group (Figure 3I,J). Meanwhile, the expression level of GFP protein in the DsRed-injected MPTP-mock group was also decreased by 43% (*n* = 5) compared to that of the negative control. In this case, the expression level of GFP was increased by 6.3, 15.8, 27.6, and 26.5 (*n* = 5) in the embryos injected with Pgk1 at 2, 4, and (2 + 4) dpf, and with Pgk1-T378P at (2 + 4) dpf, respectively (Figure 3I,K). These data suggested that the addition of Pgk1 could reverse the defects caused by the MPTP treatment in embryos such as the decrease in dopaminergic cell number by apoptosis and the decreased expression of TH and GFP proteins. Specifically, such reversal driven by Pgk1 was significantly evident in the (2 + 4)-injection group in which Pgk1 was injected twice into embryos at both 2 and 4 dpf. Importantly, the reversal of dopaminergic cell death that resulted from the addition of ePgk1 to MPTP-treated zebrafish is independent of Pgk1′s role as a glycolytic enzyme.

If we examined the number of dopaminergic neurons at the early embryonic stage, such as the 4 dpf, the average number of dopaminergic neurons in vDC of zebrafish embryos was 59.6 ± 1.49 (*n* = 6) vs. 35.6 ± 2.69 (*n* = 4) in the negative control and the MPTP-mock control treated in the manner described above, respectively (Appendix A). Meanwhile, the average number of dopaminergic neurons in the zebrafish embryos treated MPTP from 2 to 4 dpf combined with a Pgk1 injection at 2 dpf was increased to 44.6 ± 4.02 (Appendix A). Similarly, the expression level of TH protein in the DsRed-injected MPTP mock control group was significantly decreased by 58% (*n* = 5) compared to that of the untreated negative control, which was set as 1 (Appendix A). However, the expression level of TH protein was increased by 14% (*n* = 5) in the embryos injected with Pgk1 at 2 dpf compared to that of the DsRed-injected MPTP-mock control group. In the parallel experiment, the expression level of GFP in the DsRed-injected MPTP-mock group was decreased by 38% (*n* = 5) compared to that of negative control, while the expression level of GFP was increased by 19% (*n* = 5) in the embryos injected with Pgk1 at 2 dpf compared to that of the DsRed-injected MPTP mock group (Appendix A). Based on these data, we concluded that the addition of Pgk1 had a protective effect on the dopaminergic neurons exposed to MPTP from 2 to 4 dpf, such that MPTP-induced cell death was mitigated in embryos injected with Pgk1 at 2 dpf. However, we noticed that if embryos were treated with MPTP from 2 to 6 dpf combined with a Pgk1 injection once at 2 dpf, as described in previous section, the Pgk1-mediated mitigation effect of dopaminergic neurons number was not observed at 6 dpf (Figure 3E,H–K). This phenomenon might be due to the fact that the injected Pgk1 would not last in treated embryos for as long as 6 dpf.

Next, when we injected exogenous recombinant Pgk1 fused with Flag reporter into the brain chamber of 4-dpf larvae from the zebrafish transgenic line *Tg(dat:EGFP)*, we observed the location of the Pgk1-flag in the brain after 6 h, using immunostaining detection. Results showed that the Pgk1 signal was apparent on dopamine neurons and dopamine neural extended neurites (Figure 4). The line of in vivo evidence suggested that that the ePgk1 might be bound with the dopamine cell membrane. Moreover, in an in vitro system, we added ePgk1-flag into the medium cultured NSC34 motor neuron cells. Results showed that ePgk1 was also detected on the NSC34 cell membrane (Appendix A). Based on in vivo evidence, it is plausible that ePgk1 might be associated with cell membrane proteins of dopaminergic neurons and neuronal cells at the olfactory lobes. However, detailed data should be confirmed by further study in terms of location and cell types.

### 2.3. CAMI of Pgk1 Could Reduce MPTP-Induced Impaired Swimming Performance of Zebrafish

A reduced number of dopaminergic neurons led to a drop in the level of dopamine secretion, resulting in the poor swimming ability of zebrafish larvae [26]. Therefore, we further studied whether Pgk1 injected into brain chamber could improve the swimming performance of MPTP-treated larvae. The results demonstrated that the average swimming distance of zebrafish larvae in the negative control was 4.54 cm, while MPTP-treated embryos injected with DsRed protein at 2 dpf showed an average swimming distance of 1.55 cm (Figure 5). Thus, compared to the negative control, the swimming distance had decreased by about 66% in the mock group. Meanwhile, the average swimming distance of MPTP-treated embryos injected with Pgk1 protein at 2 dpf was 1.66 cm, similar to that of the mock group, having no obvious rescue effect. However, the average swimming distance of MPTP-treated embryos injected with Pgk1 protein at 4 and (2 + 4) dpf was 2.46 and 2.9 cm, respectively. Thus, compared to the DsRed-injected group, the average swimming distance of MPTP-treated embryos injected with Pgk1 protein at 4 and (2 + 4) dpf was increased by about 20 and 29%, respectively. Collectively, this line of evidence suggested that the direct injection of Pgk1 into the brain chamber rescued the MPTP-induced impairment of the swimming performance of zebrafish larvae but had a better rescue effect on embryos injected with Pgk1 at (2 + 4) dpf. Moreover, we found that embryos injected with glycolysis kinase-deficient mutant Pgk1-T378P exhibited average swimming distance similar to that of MPTP-treated embryos injected with Pgk1 at (2 + 4) dpf with an increase of about 28% in swimming distance over that of the DsRed-injected MPTP mock group. These data strengthened the hypothesis proposed in the above section that the improvement of the swimming performance of MPTP-treated zebrafish larvae by Pgk1 is independent of its intracellular canonical role of supplying metabolic energy.

We next asked if the CAMI of Pgk1 could also improve the swimming performance of MPTP-treated adult zebrafish. As shown in Figure 6A, intraperitoneal (IP) injection of MPTP was administered to adult zebrafish with a three day interval, and behavioral analysis was performed on day six after the first injection. Compared to the untreated control group, the swimming distance of the MPTP-injected adult zebrafish was decreased in a dose-dependent manner that ranged from 40 to 200 μg/g (Figure 6B) Nevertheless, the MPTP-injected adult zebrafish combined with the CAMI of Pgk1 still exhibited better swimming performance compared to fish injected with MPTP alone at any concentration (Figure 6B). Overall, the most significant improvement in swimming performance, as driven by Pgk1, was seen in the adults zebrafish treated with MPTP at the concentration of 60 and 80 μg/g. This line of evidence suggested that direct injection of Pgk1 into brain chamber alleviates the impairment of swimming performance in adult zebrafish induced by MPTP.

Finally, we employed a zebrafish transgenic strain *Tg(mnx:EGFP)*, where the motor neurons are specifically labeled by GFP, to examine whether the development of motor neurons was impeded by the exposure to MPTP from 2 to 6 dpf, resulting in poor swimming performance at 6 dpf. The results demonstrated that, compared to untreated embryos, the motor neurons of MPTP-treated *Tg(mnx:EGFP)* embryos developed normally (Appendix A). Taken together, it was concluded that the decrease of swimming ability of MPTP-treated zebrafish embryos resulted from the reduced number of dopaminergic cells, which, in turn, decreased the secretion of dopamine, not as a result of the defective motor neurons.

### 2.4. Extracellular Addition of Pgk1 Suppresses Apoptosis and ROS Levels in MPP^+^-Treated SH-SY5Y Cells

It has been reported that the overexpression of mitochondrial ROS is commonly observed in PD patients, causing apoptosis of dopaminergic neurons in brain [3,44]. We first examined the viability of SH-SY5Y human neuroblastoma cells treated with different concentrations of MPP^+^ in vitro to induce the production of ROS and cytotoxicity [45,46]. The concentration of 0.5 mM MPP^+^ was found to maintain 72% of cell survival 24 h after treatment (Appendix A). Thus, we chose 0.5 mM MPP^+^ for subsequent experiments. Next, we found that the mitigation of MPP^+^-induced cell death, as mediated by ePgk1, was dose-dependent and that 792 ng/mL Pgk1 added into 2 mL medium resulted in the better rescue of cells subjected to 0.5 mM MPP^+^ treatment (Appendix A).

We further determined whether the involvement of downstream signaling pathways of ePgk1 could prevent apoptosis and mitochondrial ROS levels induced by MPP^+^ treatment. Compared to the untreated control group, the expression level of antiapoptotic protein Bcl-2 was downregulated in the MPP^+^-treated SH-SY5Y cells, while the expression level of proapoptotic protein cleaved caspase-3 was upregulated (Figure 7A–C), suggesting that MPP^+^ treatment could induce cellular apoptosis of SH-SY5Y cells. However, when either Pgk1 or kinase-deficient mutant Pgk1-T378P was added to MPP^+^-treated SH-SY5Y cells, the expression level of Bcl-2 was upregulated, while that of cleaved caspase-3 was downregulated (Figure 7A–C), suggesting that ePgk1 could reduce MPP^+^-induced apoptosis.

We continued to study the expression level of oxidation-related proteins. Under oxidative stress, phosphorylated transcription factor nuclear factor erythroid-2-related factor 2 (p-Nrf2) could upregulate the expression of the antioxidant HO-1 gene to increase the level of HO-1 protein, causing the decrease of the ROS level [47,48]. The expression levels of p-Nrf2 and HO-1 were all significantly downregulated in the MPP^+^-treated group compared to the untreated control group (Figure 7D–F). However, these downregulated expressions of p-Nrf2 and HO-1 in the MPP^+^-treated cells could be restored by adding both ePgk1 and mutant Pgk1-T378P to the medium (Figure 7D–F), suggesting that ePgk1 increases p-Nrf2 expression and its downstream effector antioxidant HO-1, resulting in rescuing MPP^+^-induced SH-SY5Y cell death through the Nrf2/HO-1 signaling pathway.

We then utilized the MitoSOX assay to detect mitochondrial ROS generation to determine whether ROS levels were affected by the addition of Pgk1 into the culture medium of SH-SY5Y cells treated with MPP^+^. In this case, the intensity of MitoSOX red fluorescence signal induced by MPP^+^ treatment was increased three-fold compared to the untreated control group, which was set as 1 (Figure 7G vs. Figure 7H,I). However, when Pgk1 was added to the MPP^+^-treated SH-SY5Y cells, the intensity of MitoSOX red fluorescence signal was decreased by approximately two-fold over that of the untreated control group (Figure 7G vs. Figure 7I,J). These data suggest that the addition of Pgk1 could reduce the mitochondrial ROS generation induced by MPP^+^.

## 3. Discussion

Our cumulative evidence shows that extracellular administration of Pgk1 could rescue both the number of dopamine-producing cells in ventral diencephalon clusters and the swimming distances of MPTP-treated larvae and adults. These results lead to the conclusion that noncanonical ePgk1 protects dopaminergic neurons from degeneration and cell death caused by MPP^+^/MPTP-induced oxidative stress and the further speculation that it could benefit the survival of dopamine-producing cells in the PD brain, thus serving as potential neuromodulation therapy.

Zebrafish embryos treated with MPTP experience cell death in newborn dopaminergic neurons [49]. However, after MPTP-treated embryos were injected with Pgk1 via CVMI, we observed increased neuronal survival in the group of MPTP-treated embryos injected with Pgk1 once at 2 dpf, followed by once at 4 dpf, when compared to that of groups of MPTP-treated embryos injected only once with Pgk1, either at 2 dpf or 4 dpf. No such significant rescue effect on MPTP-induced cell death was observed if Pgk1 was injected into the brain of MPTP-treated embryos once at 2 dpf. While the initial BBB is developed in zebrafish embryos at 3 dpf [30], it is only incompletely formed in embryos at 2 dpf; therefore, it is reasonable to speculate that the injected Pgk1 could more freely diffuse from brain chamber to whole embryo at 2 dpf. This hypothesis is well supported by the data shown in Figure 2. Here, we showed that DsRed protein (25.9 kDa) was distributed only in the brain chamber without diffusing outward from the brain chamber at 4-dpf embryos, whereas the distribution pattern of DsRed in the 2-dpf embryos was apparent, both in the CNS and whole embryo. Our data were in complete agreement with the findings of Quiñonez-Silvero et al. [42] and Jeong et al. [43] who previously reported on the zebrafish BBB. Therefore, since the BBB is incompletely formed, the total amount of injected Pgk1 in the brain chamber of the 2-dpf recipients would be decreased by the widespread distribution of the injected Pgk1 to the trunk of the whole embryo through the blood system. In contrast, most injected Pgk1 could be retained in the brain chamber of MPTP-treated embryos at 4 dpf by the complete formation of the BBB, which leads to the significant prevention of MPTP-induced cell death of dopaminergic neurons.

### 3.1. Effect of MPTP on Dopaminergic Neurons

The catecholaminergic system in the CNS consists of catecholamine neurotransmitters, such as dopamine. Catecholaminergic neurons, including dopaminergic, adrenergic, and noradrenergic neurons, are involved in the secretion of neurotransmitters from the central and peripheral nervous systems, as well as hormones from the endocrine system [50]. The tyrosine hydroxylase (TH) uses tetrahydrobiopterin to convert tyrosine to l-DOPA. This is the initial step in the catecholamine biosynthetic pathway, and l-DOPA, the precursor to the neurotransmitters dopamine, norepinephrine (noradrenaline), and epinephrine (adrenaline), is expressed by all catecholaminergic neurons. In this case, when we observed differences in the degree of TH expression between the MPTP-treated group and the Pgk1-injected, MPTP-treated group, TH expression between these groups was not so obvious. In contrast, the dopamine transporter (DAT) is exclusively expressed in dopaminergic neurons in vertebrates [14]. When we employed the transgenic zebrafish *Tg(dat:EGFP)* [19], the GFP signal was tagged only by dopaminergic neurons, we could observe that the differences in the degree of GFP expression between the MPTP-treated group and the Pgk1-injected, MPTP-treated group were relatively obvious. To clarify, not all TH-expressing cells were damaged by MPTP; thus, the degree of TH expression would, correspondingly, be less noticeable. However, since only dopaminergic neurons were labeled by GFP in the *Tg(dat:EGFP)* embryos, differences in the degree of GFP expression was more detectable based on the all-or-none law in physiology. Therefore, we believe that this standard is sufficient and that it allows sufficient latitude by which to detect the amount of GFP expressed in *Tg(dat:EGFP)* embryos, hence the number of dopamine neurons damaged by MPTP, as reported by Kalyn and Ekker [29].

### 3.2. Extracellular Pgk1 Prevents Apoptosis Caused by Oxidative Stress in Dopaminergic Neuronal Cells

Intracellular Pgk1 is a well-known ATP supplier since it is one of the major kinases involved in glycolysis [33]. Thus, it is reasonable to think that Pgk1 deficiency would be closely related to neurodegenerative diseases, owing to defective metabolic energy production [51,52,53]. Accordingly, [51] proposed a strategy of improving glycolysis as a therapeutic target for Parkinson’s disease, by enhancing kinase activity of intracellular Pgk1 by Terazosin could promote energy production to delay the course of PD. The study of [51] reported that the climbing capability of rotenone-treated *Drosophila melanogaster* increased by approximately 15 and 20% in flies with overexpressed Pgk1 in dopaminergic neurons via the TH promoter and in flies with overexpressed Pgk1 in all neurons via pan-neuronal promoter, respectively. These data suggested that the overexpression of intracellular Pgk1 can supply more energy to oxidative stress neurons, leading to the rescue of the climbing behavioral defects of flies. 

On the other hand, extracellular Pgk1 secreted from cells may play a noncanonical function different from the glycolytic function of intracellular Pgk1. For example, tumor cells increase the expression of intracellular Pgk1 and glycolysis to facilitate energy production, but inhibit the secretion of Pgk1 in the hypoxic microenvironment. Lay et al. reported that Pgk1 secreted from human fibrosarcoma, breast, colon, and pancreatic tumor cells participates in the anti-angiogenic process through its disulphide reductase activity, which reduces the disulphide bonds in plasmin to release the tumor blood vessel inhibitor Angiostatin, which, in turn, inhibits tumor angiogenesis and growth in mice [54]. Thus, the limited secretion of Pgk1 from tumor cells would favor the angiogenesis and growth of tumors. However, Lin et al. found that NogoA-overexpressing muscle cells reduce Pgk1 secretion, resulting the inhibition of neurite outgrowth of motor neurons [38]. Thus, they demonstrated that Pgk1 secreted from muscle cells could promote neurite outgrowth of motor neurons, resulting in the alleviation of NMJ denervation and the progressive loss of motor ability in both ALS mice and ALS-like zebrafish animal models, suggesting that extracellular Pgk1 emits a cross-tissue communication signal between muscle and neuronal cells. Furthermore, similar to the injection of wild-type Pgk1, we herein demonstrated that glycolytic kinase-deficient Pgk1-T378P [55], injected into the brain of zebrafish embryos could prevent the death of dopaminergic neuronal cells caused by MPTP-induced-oxidative stress, resulting in a significant improvement of swimming ability. This line of evidence strongly supports our hypothesis that extracellular Pgk1 prevents apoptosis of dopaminergic neuronal cells in MPTP-treated embryos and adults in a manner independent of the glycolytic pathway. Additionally, based on in vivo and in vitro evidence, we hypothesized that it is plausible that ePgk1 might be associated with the cell membrane protein of dopaminergic neurons. Therefore, we propose a new hypothesis, which holds that ePgk1 can resist intracellular oxidative stress to improve the cell viability of dopaminergic neurons and the swimming ability of zebrafish treated with MPTP. However, detailed molecular mechanisms regarding how ePgk1 activates the Nrf2/HO-1 signaling pathway and contributes to the protective effects of against oxidative stress should be confirmed by further study.

### 3.3. CVMI on Zebrafish Is a Simple and Effective Approach to Study CNS Diseases

Using either a loss-of-function strategy by injecting either antisense Morpholino oligonucleotide or CRISPR/Cas9 technology to knock down or knock out Pgk1 are options to study ePgk1 function in the brain. Instead, we used a gain-of-function strategy by employing CVMI to microinject exogenous Pgk1 protein directly into the brain chamber of zebrafish. We did this because intracellular Pgk1 is involved in the glycolysis of neural cells as an energy supplier, and this results in the decreased survival of dopamine neurons through the lack of Pgk1. We can conclude that the CVMI approach performed on zebrafish embryos and adults can circumvent the problems associated with the BBB and may, therefore, provide a fast, simple, and cost-effective in vivo platform to study the physiological changes that occur in CNS diseases during the process of testing new drugs and protein/peptides. For example, using CVMI of MPTP or α-synuclein on the zebrafish brain could generate a zebrafish PD-like model [29,56]. They demonstrated that fragmented mitochondria in MPTP-injected fish could induce dopaminergic cell death activated by a mitophagy mechanism. In another example, a derivative of Aβ42 peptide with tissue penetrating capability was delivered into the adult zebrafish brain through CVMI. This resulted in generating an Alzheimer’s disease-like model fish, providing the biomaterial needed to understand how neural stem/progenitor cells are affected by neurodegenerative diseases [57,58]. In another study, CVMI of cerebrospinal fluid extracted from encephalitis patients with an anti-*N*-methyl-d-aspartate receptor (anti-NMDAR) was used to generate zebrafish displaying reduced seizure threshold and memory deficit [59].

In the present study, after we performed an CVMI of ePgk1 into the brain of MPTP-treated zebrafish, we found that dopamine cell death in the treated zebrafish, both embryos and adults, was rescued and that locomotion was improved. Additionally, the injected Pgk1 was observed around dopamine cells in the brain. Therefore, as presented in this study and supported by the reports above, we believe that the CVMI of zebrafish brains provides a convenient approach to study the efficacy of new drugs and peptides on CNS diseases in the future.

## 4. Materials and Methods

### 4.1. Plasmid Construction

The coding region of human Pgk1 cDNA (NM_000291.4) fused with FLAG-tag was inserted into a cloning vector pCS2^+^ to generate a 5.3 kb plasmid pCS2-hPgk1-flag. hPgk1-T378P, a kinase-deficient mutated form of Pgk1, was obtained by PCR-based in vitro point mutagenesis using forward primer: 5′-GGTGGAGACACTGCCCCTTGCTGTGCC-3′ and reverse primer: 5′-GGCACAGCAAGGGGCAGTGTCTCCACC-3′, in which proline (CCT) substitutes for threonine (ACT). The resultant 1.2-kb PCR product was inserted into pCS2^+^ to finally generate the 5.3-kb plasmid pCS2-hPgk1-T378P-flag.

### 4.2. Protein Expression and Purification

Protein purification followed the procedures previously described by Lin et al. [38]. Briefly, after plasmids containing cDNA of recombinant FLAG-fusion proteins were transfected into HEK293T cells, transfected cells were lysed using Pierce™ IP lysis buffer (Thermo Fisher Scientific, USA), containing protease inhibitor cocktail (Roche Applied Science, Basel, Switzerland). After cell debris was removed by centrifugation, anti-FLAG M2 affinity gel beads were added to cell extracts and incubated at 4 °C for 16 hr. The beads-FLAG-protein complex was eluted by incubation with 3× FLAG peptide for one hour. The eluate containing FLAG-fusion protein was restored and read for the next experiments. *Escherichia coli* BL21, containing plasmid pET15b-His-DsRed, could translate recombinant proteins after induction by 0.1 mM Isopropyl β-D-1-thiogalactopyranoside at 37 °C for 1 h. Thereafter, these recombinant proteins were purified through Histidine resin (GE Healthcare Biosciences AB, Uppsala, Sweden).

### 4.3. Larvae Swimming Performance

Zebrafish embryos at one day after fertilization (dpf) were incubated with Pronase (0.01 g/10 mL) for dechorionation. Then, embryos were immersed in water with 45 µM MPTP (Sigma-Aldrich, St. Louis, MO, USA, M0896) from 2 to 6 dpf, while the control group was immersed in water with DMSO from 2 to 6 dpf. Pgk1 (500 pg) was injected in a single shot into the brain of embryos either at 2 or 4 dpf. In a parallel experiment, Pgk1 was injected in a double shot into the brain, both at 2 and 4 dpf. When embryos developed into larvae at 6 dpf, the number of dopaminergic neurons located in vDC was calculated using immunostaining. Meanwhile, the swimming performance of 6-dpf larvae was also evaluated. After larvae were left to stand in a 3 cm dish for 1 min, mechanosensory stimulus was delivered to the head of each larva by touching it gently with the tip of a fishline. The route of movement was recorded by video, and swimming distance was quantified by ImageJ. The same protocol was subsequently repeated twice for each larva. The total swimming distance of each treated larva was obtained by adding the distance in cm from three mechanosensory stimuli. One datum of the total swimming distance of each trial was averaged from the total swimming distance from at least 20 treated larvae. The final data were averaged from three independent experiments and represented as mean ± S.D.

### 4.4. Cerebroventricular Microinjection (CVMI) in Adult Zebrafish

The procedure was adapted from the one previously described by Kizil and Brand [60]. Adult wild type and *Tg(dat:eGFP)* zebrafish (6-month-old) were anesthetized with 0.1 g/L tricaine. After being placed in a tricaine-moist sponge and the gills were irrigated with the same an aesthetic throughout the procedure. A small incision of 200 μm was then generated with a 30-gauge syringe (BD Biosciences, San Jose, CA, USA) in the cranium above the anterior portion of the optic tectum, to create a small access point to the telencephalon. The prepared DsRed (100 ng/μL) or Pgk1 (100 ng/μL) was co-administered with Phenol Red in PBS, than loaded into a thin microinjecting glass needle to inject 0.2 μL of the contents through the incision into the cerebroventricular fluid surrounding the brain.

### 4.5. Western Blotting

Western blot analysis on SDS-PAGE followed the procedures previously described by Lin et al. [61], except that the antibodies against tyrosine hydroxylase (TH) (ImmunoStar LD; Fujifilm-Wako, Osaka, Japan, 22941; 1:1000), GFP (Abcam, Cambridge, UK, ab13970; 1:2000), α-tubulin (Sigma-Aldrich St. Louis, MO, USA, T5168; 1:5000), Flag (Abcam, Cambridge, UK, ab21536; 1:5000), goat anti-mouse-HRP (Abcam, Cambridge, UK, ab6789; 1:5000), and goat anti-rabbit-HRP (Cell Signaling Technology, Danvers, MA, USA, 70743; 1:5000) were used.

### 4.6. Immunostaining of Zebrafish Embryos

The immunostaining of zebrafish embryos was previously described by Lin et al. [38], except that the primary antibody, such as tyrosine hydroxylase (TH) (ImmunoStar LD; Fujifilm-Wako, Osaka, Japan, 22941; 1:300), GFP (Abcam, Cambridge, UK, ab13970; 1:500), Flag (Abcam, ab21536; 1:500) and secondary antibody, such as DyLight™ 488-conjugated Goat anti-mouse IgG (Rockland, Limerick, PA, USA, 610-141-121; 1:500) and goat anti-rabbit-IgG (Cell Signaling Technology, Danvers, MA, USA, 70743; 1:500), were used. The expression patterns were observed under a laser scanning confocal fluorescence microscope (Zeiss, Oberkochen, Germany).

### 4.7. Cell Culture

The human neuroblastoma cell line SH-SY5Y was cultured in DMEM/F12 medium (Gibco-BRL, Waltham, MA, USA) supplemented with 10% fetal bovine serum (Gibco-BRL, Waltham, MA, USA), 100 U/mL penicillin and 100 mg/mL streptomycin (Gibco-BRL, Waltham, MA, USA), 0.1 mM nonessential amino acids (Sigma-Aldrich St. Louis, MO, USA) and 1 mM sodium pyruvate (Sigma-Aldrich St. Louis, MO, USA), followed by incubation at 37 °C with a humidified atmosphere of 5% CO_2_. SH-SY5Y cells were then cultured in a T75 flask until grown to around 70–80% confluency. They were then subcultured into 6- or 96-well plates for subsequent experiments.

### 4.8. ROS Generation Detection

We used the mitochondria-sensitive dye MitoSOX Red for the detection of mitochondrial ROS production (Thermo Scientific, Waltham, MA, USA). SH-SY5Y cells were seeded at 2 × 10^5^ cells, treated with MPP^+^ for 24 h in the absence or presence of Pgk1 and incubated with 5 μM MitoSOX Red for 10 min. MitoSOX Red analysis was performed at 510-nm excitation and 580-nm emission.

### 4.9. Statistical Analyses

GraphPad Prism (ver. 5; GraphPad Software Inc., San Diego, CA, USA) was used to perform all statistical analyses. One-way ANOVA was used to analyze different concentrations of MPTP on the swimming ability of zebrafish larvae. Statistical significance was determined using Student’s *t*-test for parametric data. The results are shown as the mean ± SEM and a level of *p* ≤ 0.05 was considered to be statistically significant. The sample sizes used in each experiment are mentioned in the results section and figure legends.

## Figures and Tables

**Figure 1 ijms-23-04150-f001:**
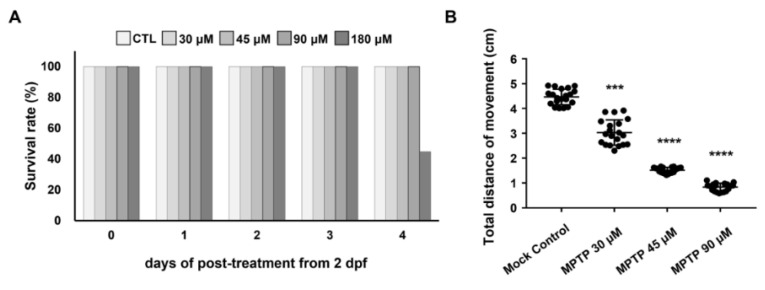
Effect of different concentrations of MPTP on the survival rate and swimming ability of zebrafish larvae. (**A**) The daily post-treatment survival rate of embryos immersed with different concentrations of MPTP, as indicated, starting at 2 dpf for 4 days. (**B**) Statistical analysis of the total swimming distance of larvae treated with different concentrations of MPTP as indicated. Larvae cultured in a 3 cm dish and treated with DMSO served as a mock control. The total distance (in cm) of a single larva was the sum of its swimming distance instigated by touches to the head and was averaged from three trials. The total distance of each experiment was the average total distance obtained from 20 larvae, while that of each group was averaged from three independent experiments. Student’s *t*-test was used to determine significant differences between each group (****, *p* < 0.0001; ***, *p* < 0.001).

**Figure 2 ijms-23-04150-f002:**
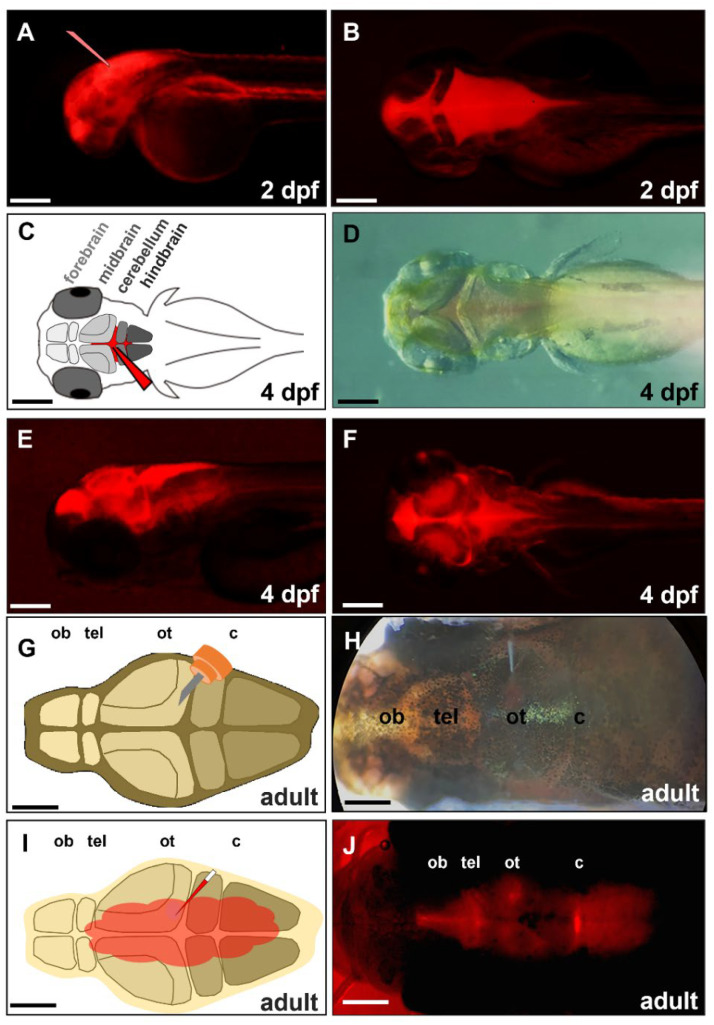
Distribution of the injected exogenous DsRed protein in brain chamber. We injected exogenous red fluorescence protein (DsRed) into the brain chamber of a zebrafish embryo at (**A**,**B**) 2-, (**C**–**F**) 4-dpf, and (**G**–**J**) a 6-month-old adult zebrafish. (**A**) The glass needle represents the injection site of zebrafish embryo’s head at 2 dpf. (**C**) Schematic diagram depicts the direct injection of DsRed into the brain chamber of zebrafish embryo at 4 dpf. The glass needle marked in red represents the injection site. (**D**) Dorsal view of 4-dpf embryo injected with DsRed under bright-field microscopy; (**E**) Lateral view of 4-dpf embryo injected DsRed under fluorescence microscopy; (**F**) Dorsal view of 4-dpf embryo injected DsRed under fluorescence microscopy. (**G**) Schematic diagram depicts a small incision that was generated above the optic tectum of adult zebrafish by a 30-gauge syringe. (**H**) Dorsal view of adult zebrafish injected DsRed by glass needle under bright-field microscopy; (**I**) Schematic diagram depicts the distribution of injected DsRed in the brain through the cerebroventricular fluid. (**J**) Dorsal view of adult zebrafish injected DsRed after 30 min under fluorescence microscopy. ob: olfactory bulb, tel: telencephalon, ot: optic tectum, c: cerebellum. Scale bar: (**A**–**F**) 25 µm; (**G**–**J**) 500 µm.

**Figure 3 ijms-23-04150-f003:**
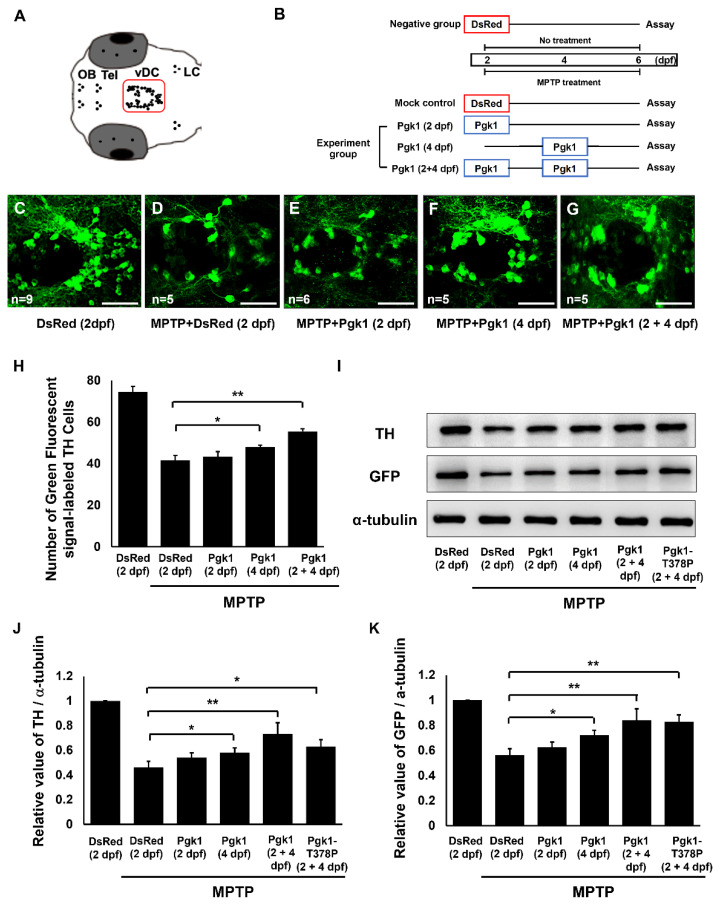
Direct injection of Pgk1 into the brain chamber prevents MPTP-induced dopaminergic cell death in the ventral diencephalon of zebrafish embryo at 6 dpf. (**A**) Schematic diagram depicts the distribution of dopaminergic neuronal cells in the brain of zebrafish embryo. The vDC region is marked with a square. (**B**) Schedule of MPTP treatment for embryos from 2 to 6 dpf. Negative control: embryos were injected with DsRed protein at 2 dpf without MPTP treatment. Mock control: embryos were injected with DsRed protein at 2 dpf and treated with 45 µM MPTP from 2 through 6 dpf. Experimental groups: Pgk1 was individually injected at 2, 4, and 2 plus 4 (2 + 4) dpf and then treated with MPTP from 2 to 6 dpf. Immunostaining assay was performed at 6 dpf. (**C**–**G**) Immunostaining of zebrafish embryos at 6 dpf. Using confocal microscopy, Tyrosine hydroxylase (TH)-specific antibody, labeled with a green fluorescence signal, could be observed in the vDC region of embryos with different treatments, as indicated. (**H**) Statistical analysis of the average number of the green fluorescent signal-labeled TH cells located in vDC. Number of examined embryos was indicated at the lower left corner of each panel. Projections of Z-stack images were generated with 2 µm. (**I**) Western blot analysis of the TH and GFP proteins expressed in the head of 10 larvae from transgenic line Tg(*dat:EGFP*) treated as indicated. The α-tubulin served as an internal control. The relative expression values of (**J**) TH and (**K**) GFP quantified from different groups after normalization of the expression level of α-tubulin. The level of each examined protein in the negative group set was expressed as 1. All data were averaged from three independent experiments and represented as mean ± S.D. Student’s *t*-test was used to determine significant differences between each group (**, *p* < 0.01; *, *p* < 0.05). OB: olfactory bulb; Tel: telencephalon; vDC: ventral diencephalon; LC: locus coeruleus. Scale bar, 50 μm.

**Figure 4 ijms-23-04150-f004:**
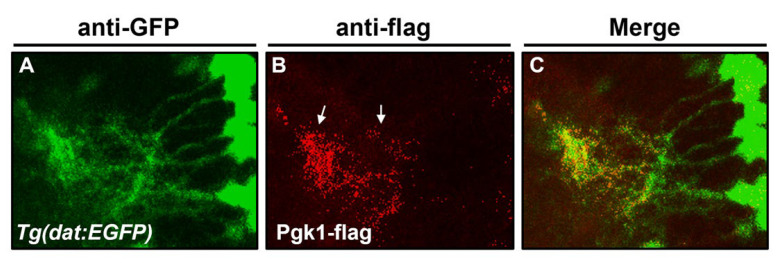
The injected Pgk1 was located at the surface of dopamine cells in the brain chamber. Using immunofluorescent staining to detect the spatial distribution of dopamine neurons and the location of injected Pgk1 in the zebrafish embryos at 4 dpf. (**A**) The dopamine neurons of transgenic line *Tg(dat:EGFP)* were labeled by anti-GFP antibody conjugated with green signals. (**B**) The exogenously injected Pgk1-flag was labeled by the anti-flag antibody conjugated with red. (**C**) The red-labeled Pgk1-Flag located with green-labeled dopamine neurons and neurites was indicated by white arrows.

**Figure 5 ijms-23-04150-f005:**
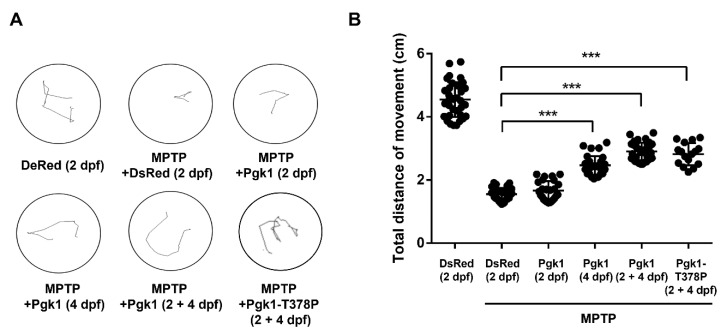
Injection of recombinant Pgk1-Flag into the brain could partially rescue the impairment of swimming performance of MPTP-induced zebrafish larvae. Six groups of zebrafish larvae at 6 dpf, as indicated, were categorized: (1) DsRed (2dpf): embryos at 2 dpf were injected with DsRed protein without MPTP treatment and served as a negative control; (2) MPTP + DsRed (2dpf): embryos at 2 dpf were injected with DsRed protein and treated with MPTP as a mock control; (3–5) Experimental groups: (3) MPTP + Pgk1 (2 dpf): embryos at 2 dpf were injected with Pgk1 and treated with MPTP from 2 to 6 dpf; (4) MPTP + Pgk1 (4 dpf): embryos at 4 dpf were injected with Pgk1 and treated with MPTP from 2 to 6 dpf; and (5) MPTP + Pgk1 (2 + 4 dpf): embryos at 2 and 4 dpf were individually injected with Pgk1 and treated with MPTP from 2 to 6 dpf; and (6) MPTP + Pgk1-T378P (2 + 4 dpf): embryos at 2 and 4 dpf were individually injected with mutated Pgk1-T378P and treated with MPTP from 2 to 6 dpf. The swimming route of each zebrafish larva at 6 dpf was tracked using ImageJ software, and its swimming distance was calculated in cm. (**A**) The three times head touch-evoked swimming routes of a single larva in six groups of zebrafish larvae were recorded. (**B**) Statistical analysis of the total swimming distance (in cm) per larva in each group. The total distance of a single larva was the sum of swimming distance instigated by touches to the head and averaged from three trials. The total distance of each experiment was the average total distance obtained from 20 to 35 larvae, while that of each group was averaged from three independent experiments. Student’s *t*-test was used to determine significant differences between each group (***, *p* < 0.001).

**Figure 6 ijms-23-04150-f006:**
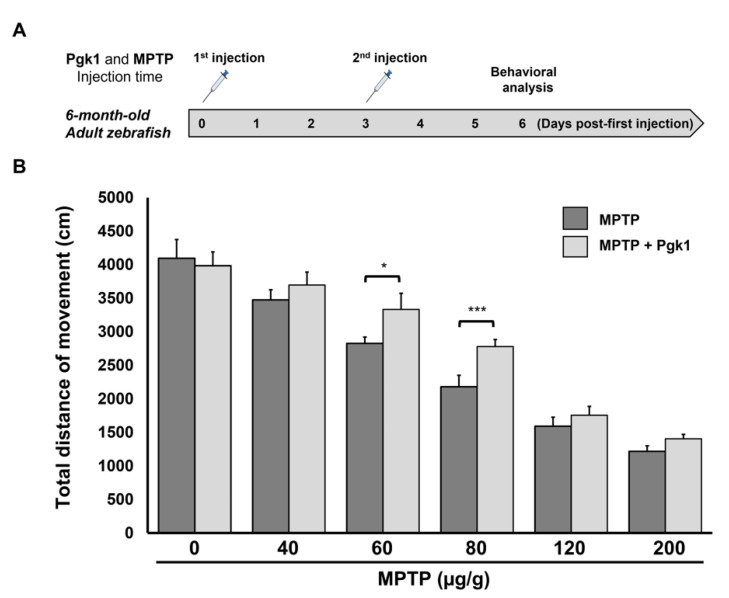
Direct injection of Pgk1 into the brain of adult zebrafish alleviates the impairment of swimming performance induced by MPTP. (**A**) Schematic diagram to illustrate the timetable of the six-month-old adult zebrafish being treated with intraperitoneal injections of MPTP, combined with a brain injection of Pgk1, followed by behavioral analysis. (**B**) Statistical analysis of total swimming distance (in cm) of adult fish treated with different concentrations of MPTP, as indicated, combined with, or without, a Pgk1 injection. The total swimming distance was averaged from three independent trials, and the distance of each trial was averaged from five adult fish, while the distance of a single fish sample was averaged from three touches. Student’s *t*-test was used to determine significant differences between each group (***, *p* < 0.001; *, *p* < 0.05).

**Figure 7 ijms-23-04150-f007:**
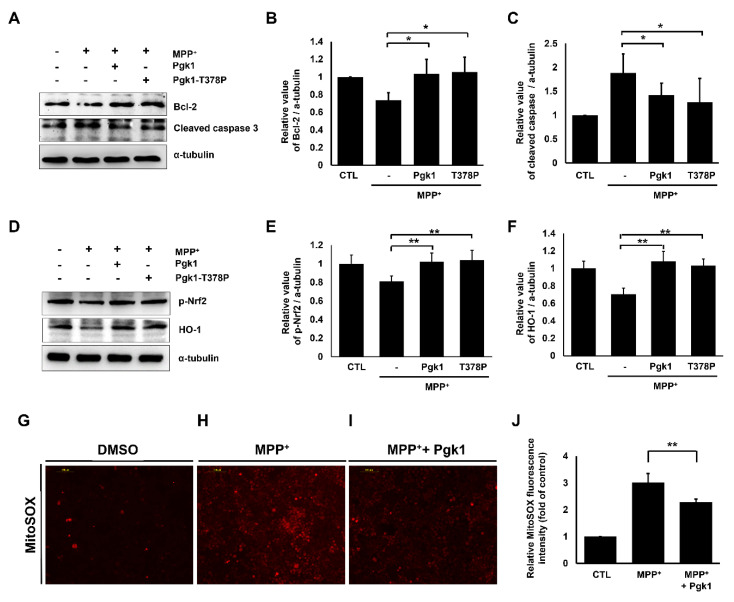
Extracellular addition of Pgk1 suppresses MPP^+^-induced apoptosis and mitochondrial ROS levels in SH-SY5Y cells. (**A**,**D**) Western blot analysis. The expression levels of Bcl-2, cleaved caspase-3, and p-Nrf2 and HO-1 proteins, as indicated, of SH-SY5Y cells treated or not treated with 0.5 mM MPP^+^ in the absence or presence of Pgk1 or kinase-deficient mutant T378P Pgk1 for 24 h. The α-tubulin served as an internal control of loading protein. (**B**,**C**,**E**,**F**) Statistical analysis of relative value of target protein expressed in cells with different treatment after normalization of the expression level of α-tubulin. (**G**–**I**) The microscopic images of MitoSOX red fluorescence signal shown on cells. SH-SY5Y cells treated with or without 0.5 mM MPP^+^ in the absence or presence of Pgk1 and then stained with MitoSOX to detect the level of ROS generated in cells. (**J**) Quantification of intensity of fold increase of MitoSOX red fluorescence signal compared to that of cells without treatment served as a control group (CTL), which was set as 1. All data were averaged from three independent experiments and represented as mean ± S.D. Student’s *t*-test was used to determine significant differences between each group (**, *p* < 0.01; *, *p* < 0.05).

## Data Availability

The datasets analyzed during the current study are available from the corresponding author on reasonable request.

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
