# Peer review of "Cerebroventricular Injection of Pgk1 Attenuates MPTP-Induced Neuronal Toxicity in Dopaminergic Cells in Zebrafish Brain in a Glycolysis-Independent Manner"

_ijms, 2022, doi:10.3390/ijms23084150_

Round 1

Reviewer 1 Report

In this manuscript, Lin and colleagues treated zebrafish (at various life stages) with MPTP and ePGK and show a neuroprotective effect of the latter, effect that is independent of the enzymatic activity of PGK in glycolysis. This study follows another one from the same group published in 2019 that showed an effect of ePGK on neurite outgrowth in motoneurons.

Overall, the current work could be interesting but the overall story relies on relatively weak evidence that do not offer a convincing mechanistic insight. Additional work is necessary to make it suitable for publication.

Finally, the writing of the manuscript requires major work. In addition to the minor but numerous problems with the English, there are still quite a few sentences that are unclear as written, as outlined below.

Major points:

Figure 2: after figuring out what was the purpose of this experiment in the context of the manuscript (see my comment below), it is this reviewer’s opinion that the evidence presented here is too preliminary/unconvincing to support the conclusions made by the authors in their Discussion (lines 400-413). The experiment described in Figure 2 is rather crude and images shown are moderately convincing. Additional experiments should be made to directly address the movement, or lack thereof, of proteins across the BBB, in both directions, to adequately support the conclusions.

Apoptosis: The authors address possible impact of MPTP/MPP on apoptosis and the protective effects of PGK1. However, they only provide indirect evidence and never looked at apoptosis. This should be done or the entire set of results should be removed or addressed with a lot more caution.  In this respect, the decreases in Bcl2 levels after MPP+ shown in Figure 7B are modest. Do the authors know, perhaps based on other work, that such decreases are sufficient to affect apoptosis? If not, a mention to the fact that “this decrease is modest and it is not known if could affect apoptosis” should be made.

Minor points:

Figure 2: There are several problems with this figure and its description, in addition to the one listed under “Major comments”: First, it took me some time to figure out exactly why the authors were doing this experiment. It was clearer once I read the Discussion. In the section of Results that deals with Figure 2, the authors should start by saying what is the purpose of the experiment. The conclusion should tell us why their conclusion is important for what follows in the manuscript. Two additional points: a) panel 2J is not mentioned in text; b) the sentence at lines 392-393 of the Discussion, standing there by itself, is out of place. It should either be merged with the rest of the Discussion sentences that address the BBB or be used as the title of a sub-section of the Discussion.

Line 250-251: The binding of PGK1 to the cell membrane of dopaminergic neurons. Does it apply to other cells in the brain? If the authors did not examine this, a mention to this effect should be made.

Line342-344: the causal links with ROS that are implied are too strong, especially that ePGK1 is thought to act outside the cell and ROS are inside. This should be thoroughly rephrased with more caution. The same applies to the sentence on lines 471-473, where the logical link/mechanism is unclear to this reviewer.

Line 391-392 The statement about neuromodulation therapy is a long stretch as it avoids discussion of the difficulties of using/delivering ePGK for that purpose. This should be carefully thought or deleted.

The paragraph on lines 501-507 does not work. It tries to link conclusions from different stories that may not even be connected.

Minor points about writing:

Line 88 : Remove the word « Consequently,”

Line 248: remove the word “culturing”

Line 330 Remove the word “were”

Line 333-334: “dopamlamine” What is this? Please clarify where this term comes from.

Line415-418 This sentence is either unclear as written or inaccurate, especially the part about “mediate the effects of harmful neurotoxins”. Some rewriting is in order.

The text on lines 431-435 is totally unclear to this reviewer.

Line 492-493. This sentence is unclear to me and I am not even sure how to change it.

Author Response

Reviewer 1

In this manuscript, Lin and colleagues treated zebrafish (at various life stages) with MPTP and ePGK and show a neuroprotective effect of the latter, effect that is independent of the enzymatic activity of PGK in glycolysis. This study follows another one from the same group published in 2019 that showed an effect of ePGK on neurite outgrowth in motoneurons.

Overall, the current work could be interesting but the overall story relies on relatively weak evidence that do not offer a convincing mechanistic insight. Additional work is necessary to make it suitable for publication.

Finally, the writing of the manuscript requires major work. In addition to the minor but numerous problems with the English, there are still quite a few sentences that are unclear as written, as outlined below.

A. Major points:

A1. Figure 2: after figuring out what was the purpose of this experiment in the context of the manuscript (see my comment below), it is this reviewer’s opinion that the evidence presented here is too preliminary/unconvincing to support the conclusions made by the authors in their Discussion (lines 400-413). The experiment described in Figure 2 is rather crude and images shown are moderately convincing. Additional experiments should be made to directly address the movement, or lack thereof, of proteins across the BBB, in both directions, to adequately support the conclusions.

Author’s response to A1:

In response to your comment, we rewrote the caption of Figure 2 to emphasize the importance of using CAMI for zebrafish to study the effect of protein on the developing central nervous system (CNS), as explained in the Results section: “Under in vivo systems, the blood-brain barrier (BBB) poses numerous obstacles to the study of drug/protein/peptide efficacy against such CNS diseases as brain tumor, Parkinson’s disease and Alzheimer’s disease in the brain of mammals [40]. The structure and function of zebrafish BBB are reported to be similar to those of mammals [41]. Moreover, Quiñonez-Silvero et al. [42] reported that the zebrafish BBB begins to develop hindbrain vascularization in embryos at 28-32 hours post-fertilization (hpf). After 72 hpf, the permeability of the vessels is quite limited owing to the BBB. For example, DAPI (350 Da), Rhodamine-Dextran (10 kDa) and horseradish peroxidase (HRP, 44 kDa) are impermeable to cerebral microvessels in zebrafish embryos [43]. Since the development and function of zebrafish BBB are well documented, we had to employ CVMI on zebrafish embryos and adults to circumvent such problems as those noted above. Nevertheless, before we could determine if exogenous Pgk1 was present in the brain chamber after CAMI, it was necessary to briefly perform the movement of protein across the BBB using the red fluorescence protein (DsRed), which served as a test material in this study. After injection of DsRed into brain chamber, the pattern of its fluorescence signal was individually observed in 2- and 4-dpf embryos, as well as six-month-old adult zebrafish samples (Figure 2). The distribution pattern of DsRed in the 2-dpf embryos was apparent at CNS and the body close to the embryo head (Figure 2A,B). However, DsRed was distributed throughout the whole embryos after injection for one hr. In contrast, it was only dispersed in the brain chamber in 4-dpf embryos (Figure 2C-F). Meanwhile, in a parallel experiment, we created a small incision above the anterior portion of the optic tectum of six-month-old adult zebrafish without damaging the brain (Figure 2G). After injecting DsRed into the cerebroventricular fluid surrounding the brain (Figure 2H,I), the red fluorescent signal was also only distributed evenly in the brain chamber 30 min after injection (Figure 2J). Thus, since DsRed appeared at CNS and whole body of 2-dpf embryos, these results could only be explained by incomplete formation of the BBB.” (Please see lines 148-173)

Additionally, we revised the Discussion section as follows: “Using either a loss-of-function strategy by injecting either antisense Morpholino oligonucleotide or CRISPR/Cas9 technology to knock down or knock out Pgk1 are options to study ePgk1 function in the brain. Instead, we used a gain-of-function strategy by employing CVMI to microinject exogenous Pgk1 protein directly into the brain chamber of zebrafish. We did this because intracellular Pgk1 is involved in the glycolysis of neural cells as an energy supplier, and this results in the decreased survival of dopamine neurons by the lack of Pgk1.“  (Please see lines 515-521)

“Zebrafish embryos treated with MPTP experience cell death in newborn dopaminergic neurons. However, after MPTP-treated embryos were injected with Pgk1 via CVMI, we observed increased neuronal survival in the group of MPTP-treated embryos injected with Pgk1 once at 2-dpf, followed by once at 4-dpf, when compared to that of groups of MPTP-treated embryos injected only once with Pgk1, either at 2-dpf or 4-dpf. No such significant rescue effect on MPTP-induced cell death was observed if Pgk1 was injected into the brain of MPTP-treated embryos once at 2 dpf. While the initial BBB is developed in zebrafish embryos at 3-dpf [30], it is only incompletely formed in embryos at 2 dpf; therefore, it is reasonable to speculate that the injected Pgk1 could more freely diffuse from brain chamber to whole embryo at 2 dpf. This hypothesis is well supported by the data shown in Figure 2. Here, we showed that DsRed protein (25.9 kDa) was distributed only in the brain chamber without diffusing outward from the brain chamber at 4-dpf embryos, whereas the distribution pattern of DsRed in the 2-dpf embryos was apparent, both in the CNS and whole embryo. Our data were in complete agreement with the findings of Quiñonez-Silvero et al. [42] and Jeong et al. [43] who previously reported on the zebrafish BBB. Therefore, since the BBB is incompletely formed, the total amount of injected Pgk1 in the brain chamber of 2-dpf recipients would be decreased by the widespread distribution of the injected Pgk1 to the trunk of whole embryos through the blood system. In contrast, most injected Pgk1 could be retained in the brain chamber of MPTP-treated embryos at 4-dpf by the complete formation of BBB, which leads to significant prevention of MPTP-induced cell death of dopaminergic neurons. “(Please see lines 425-445)

A2. Apoptosis: The authors address possible impact of MPTP/MPP on apoptosis and the protective effects of PGK1. However, they only provide indirect evidence and never looked at apoptosis. This should be done or the entire set of results should be removed or addressed with a lot more caution. In this respect, the decreases in Bcl2 levels after MPP+ shown in Figure 7B are modest. Do the authors know, perhaps based on other work, that such decreases are sufficient to affect apoptosis? If not, a mention to the fact that “this decrease is modest and it is not known if could affect apoptosis” should be made.

Author’s response to A2:

Üstündağ et al. [49] (2022; https://doi.org/10.1080/01480545.2020.1795189) have clearly demonstrated that the decrease in the number of dopamine cells in the brain of MPTP-treated zebrafish embryos results from the induction of a p53-dependent and Bax-mediated apoptosis in zebrafish embryos.

Gong et al. [45] (2017; Fig. 3B; https://doi.org/10.3892/etm.2017.5049) and Wang et al. [46] (2018; Fig. 3E; https://doi.org/10.1002/cbin. 10864) have clearly demonstrated that SH-SY5Y cells treated under the same conditions as those in the present study, such as MPP+ (0.5 mM) treatment for 24 hr, would induce apoptosis and oxidative stress, resulting in a decreased level of Bcl2. Moreover, they both pointed out that the Bcl2 level could serve as an indicator of MPP+-induced cytotoxicity and apoptosis. Therefore, based on the conclusions provided by Gong et al. [45] and Wang et al. [46], we specifically aimed to demonstrate the decreased level of Bcl2 in the MPP+-treated SH-SY5Y cells, as shown in Figure 7B.

B. Minor points:

B1. Figure 2: There are several problems with this figure and its description, in addition to the one listed under “Major comments”: First, it took me some time to figure out exactly why the authors were doing this experiment. It was clearer once I read the Discussion. In the section of Results that deals with Figure 2, the authors should start by saying what is the purpose of the experiment. The conclusion should tell us why their conclusion is important for what follows in the manuscript.

Author’s response to B1:

Thank you. In response to your suggestion, we revised the Results section as it pertains to Figure 2 by stating the purpose of the experiment. (Please see lines 148-173)

B2. Two additional points: a) panel 2J is not mentioned in text; b) the sentence at lines 392-393 of the Discussion, standing there by itself, is out of place. It should either be merged with the rest of the Discussion sentences that address the BBB or be used as the title of a sub-section of the Discussion.

Author’s response to B2:

(a) Thank you. We added the description of panel 2J in the text. (Please see lines 170-172)

(b) We removed the sentence from lines 392-393 and incorporated it into the Discussion section, as suggested.

B3. Line 250-251: The binding of PGK1 to the cell membrane of dopaminergic neurons. Does it apply to other cells in the brain? If the authors did not examine this, a mention to this effect should be made.

Author’s response to B3:

In response to your comments, we revised the description as follows: “Based on in vivo evidence, it is plausible that ePgk1 might be associated with cell membrane protein of dopaminergic neurons and neuronal cells at the olfactory lobes. However, detailed data should be confirmed by further study in terms of location and cell types. “ (Please see lines 277-280)

B4. Line 342-344: the causal links with ROS that are implied are too strong, especially that ePGK1 is thought to act outside the cell and ROS are inside. This should be thoroughly rephrased with more caution. The same applies to the sentence on lines 471-473, where the logical link/mechanism is unclear to this reviewer.

Author’s response to B4:

In response to your comments, we (I) added three new figures (D-F) into Figure 7, demonstrating that ePgk1 could rescue MPP+-induced SH-SY5Y cell death through the Nrf2/HO-1 signaling pathway; (II) added a new Supplementary Figure 2 to demonstrate that ePgk1 might be located at the cell membrane of neuronal cells; and (III) revised subsection of 2.4., entitled “Extracellular addition of Pgk1 suppresses apoptosis and ROS levels in MPP+-treated SH-SY5Y cells.”

(I) New Figure 7 demonstrating that ePgk1 can rescue MPP+-induced SH-SY5Y cell death through the Nrf2/HO-1 signaling pathway

Figure 7. Extracellular addition of Pgk1 suppresses MPP+-induced apoptosis and mitochondrial ROS level in SH-SY5Y cells. (A, D) Western blot analysis. The expression levels of Bcl-2, cleaved caspase-3, and p-Nrf2 and HO-1 proteins, as indicated, of SH-SY5Y cells treated with or without 0.5 mM MPP+ in the absence or presence of Pgk1 or kinase-deficient mutant T378P Pgk1 for 24 h. The α-tubulin served as an internal control of loading protein. (B-C, E-F) Statistical analysis of relative value of target protein expressed in cells with different treatment after normalization of the expression level of α-tubulin. (G-I) Microscopic images of MitoSOX red fluorescent signal shown on cells. SH-SY5Y cells treated with or without 0.5 mM MPP+ in the absence or presence of Pgk1 and then stained with MitoSOX to detect the level of ROS generated in cells. (J) Quantification of intensity of fold increase of MitoSOX red fluorescent signal compared to that of cells without treatment served as a control group (CTL), which was set as 1. All data were averaged from three independent experiments and represented as mean±S.D. Student’s t-test was used to determine significant differences between each group (***, p<0.001; **, p<0.01; *, p<0.05).

(II) New Supplementary Figure 2 demonstrating that extracellular addition of Pgk1 was located at the cell membrane of neural cells

Supplementary Figure 2: Extracellular addition of Pgk1 was located at the cell membrane of neural cells. The green fluorescent signal was used to label Na+/K+-ATPase (membrane marker; positive control), while the red fluorescent signal was used to label DsRed-Flag (negative control). (A-F) The culture medium of NSC34 was added with 33 ng/μl DsRed-Flag. No red fluorescent signal located at the neural cell membrane was observed. (G-L) The culture medium of NSC34 was added with 33 ng/μl Pgk1-Flag. The red fluorescent signal located at neural cell membrane was colocalized with green fluorescent-labeled Na+/K+-ATPase (indicated by white arrowheads), suggesting that ePgk1 may be associated with a membrane protein of neural cells.

(III) Revised subsection 2.3., entitled “Extracellular addition of Pgk1 suppresses apoptosis and ROS levels in MPP+-treated SH-SY5Y cells.”

It has been reported that the overexpression of mitochondrial ROS is commonly observed in PD patients, causing apoptosis of dopaminergic neurons in brain [3,44]. We first examined the viability of SH-SY5Y human neuroblastoma cells treated with different concentrations of MPP+ in vitro to induce the production of ROS and cytotoxicity [45,46]. The concentration of 0.5 mM MPP+ was found to maintain 72% of cell survival 24 hr after treatment (Supplemental Figure 4A). Thus, we chose 0.5 mM MPP+ for subsequent experiments. Next, we found that the mitigation of MPP+-induced cell death, as mediated by ePgk1, was dose-dependent and that 792 ng/ml Pgk1 added into 2 ml medium resulted in better rescue of cells subjected to 0.5 mM MPP+ treatment (Supplemental Figure 4B).

We further determined whether the involvement of downstream signaling pathways of ePgk1 could prevent apoptosis and mitochondrial ROS levels induced by MPP+ treatment. Compared to the untreated control group, the expression level of antiapoptotic protein Bcl-2 was downregulated in the MPP+-treated SH-SY5Y cells, while the expression level of proapoptotic protein cleaved caspase-3 was upregulated (Figure 7A-C), suggesting that MPP+ treatment could induce cellular apoptosis of SH-SY5Y cells. However, when either Pgk1 or kinase-deficient mutant Pgk1-T378P was added to MPP+-treated SH-SY5Y cells, the expression level of Bcl-2 was upregulated, while that of cleaved caspase-3 was downregulated (Figure 7A-C), suggesting that ePgk1 could reduce MPP+-induced apoptosis.

We continued to study the expression level of oxidation-related proteins. Under oxidative stress, phosphorylated transcription factor nuclear factor erythroid-2-related factor 2 (p-Nrf2) could upregulate the expression of antioxidant HO-1 gene to increase the level of HO-1 protein, causing the decrease of ROS level [47,48]. The expression levels of p-Nrf2 and HO-1 were all significantly downregulated in the MPP+-treated group compared to the untreated control group (Figure 7D-F). However, these downregulated expressions of p-Nrf2 and HO-1 in the MPP+-treated cells could be restored by adding both ePgk1 and mutant Pgk1-T378P in the medium (Figure 7D-F), suggesting that ePgk1 increases p-Nrf2 expression and its downstream effector antioxidant HO-1, resulting in rescuing MPP+-induced SH-SY5Y cell death through the Nrf2/HO-1 signaling pathway.

We then utilized the MitoSOX assay to detect mitochondrial ROS generation to determine whether ROS expression was affected by addition of Pgk1 into the culture medium of SH-SY5Y cells treated with MPP+. In this case, the intensity of MitoSOX red fluorescent signal induced by MPP+ treatment was increased three-fold compared to the untreated control group, which was set as 1 (Figure 7G vs. H, J). However, when Pgk1 was added to the MPP+-treated SH-SY5Y cells, the intensity of MitoSOX red fluorescent signal was decreased by approximately two-fold over that of the untreated control group (Figure 7G vs. I, J). These data suggested that the addition of ePgk1 could reduce the mitochondrial ROS level induced by neurotoxic MPP+. (Please see lines 362-416)

B5. Line 391-392 The statement about neuromodulation therapy is a long stretch as it avoids discussion of the difficulties of using/delivering ePGK for that purpose. This should be carefully thought or deleted.

Author’s response to B5: Thank you for your suggestion. We deleted this statement.

B6. The paragraph on lines 501-507 does not work. It tries to link conclusions from different stories that may not even be connected.

Author’s response to B6: In response to your comments, we also deleted the paragraph in lines 501-507.

C. Minor points about writing:

C1. Line 88 : Remove the word « Consequently,”

Author’s response to C1: We removed it.  

C2. Line 248: remove the word “culturing”

Author’s response to C2: We removed it.

C3. Line 330 Remove the word “were”

Author’s response to C3: We removed it.

C4. Line 333-334: “dopamlamine” What is this? Please clarify where this term comes from.

Author’s response to C4: We corrected this word to dopamine. Thank you.

C5. Line 415-418 This sentence is either unclear as written or inaccurate, especially the part about mediate the effects of harmful neurotoxins. Some rewriting is in order.

Author’s response to C5:

Thank you for pointing out this incorrect statement in lines 415-418. We corrected this sentence as follows: “The catecholaminergic system in the CNS consists of catecholamine neurotransmitters, such as dopamine. Catecholaminergic neurons, including dopaminergic, adrenergic and noradrenergic neurons, are involved in secretion of neurotransmitters from the central and peripheral nervous systems, as well as hormones from the endocrine system [50]”. (Please see lines 447-451)

C6. Line 492-493. This sentence is unclear to me and I am not even sure how to change it.

Author’s response to C6: We removed this sentence.

Reviewer 2 Report

Major comments  
  1. The rationale behind providing exogenous Pgk1 protein in the extracellular area is to mimic secretion and the internalization of Pgk1 and study its potential non-canonical functions. However, this rationale is not clear enough in the introduction section of the manuscript. The authors should justify clearly why the exogenous administration of Pgk1 is relevant.
  2. The colocalization of Pgk1 with TH antibody is not apparent. The Pgk1-Flag seems to be localized in the extracellular space rather than the cytoplasm of DA neurons. The images (Figure 4-A, B, C) are not resolved enough to confirm the colocalization. The signal from dat: EGFP is overexposed. The marked areas, especially the dotted circle, lack any specific signal! The authors need to provide better-quality images to support their data. 
  3. The authors have also not provided evidence that ePkg1 is readily absorbed into the SH-SY5Y cells without any protein penetration strategies.
  4. The authors have used Students- T-tests in their statistical analysis. The Student t-test is used to compare the means between two groups, whereas ANOVA should be used to compare the means among three or more groups. The materials and methods section lacks a subsection on statistics. The authors should describe the parameters used to employ individual statistical tests and explain this in the materials and methods section.
  5. The authors have not referenced the statement (Sentence 261) that DA neuronal loss leads to poor swimming ability. Dopamine acts as an excitatory and inhibitory neurotransmitter with contradicting effects on motility.
  Minor corrections
  1. The result section needs to be checked for grammatical errors.
  2. Figure 7 A: The alpha-tubulin band does not correspond to the Bcl-2 band in figure 1.
  3. Sentence 18: Time, not times
  4. Sentence 31: cell death of dopaminergic neurons- death of dopaminergic neurons or dopaminergic neuronal death
  5. Sentence 31 substantia nigra pars compacta in italics.
  6. Sentence 78: PD- like Zebrafish model not PD-like zebrafish.
  7. ROS expression: Reactive oxygen species levels, not expression. Expression is associated with gene or protein levels.
  8. MPP+ or MPTP treatment in SH-SY5Y cells?

Author Response

Reviewer 2

Major comments 

  1. The rationale behind providing exogenous Pgk1 protein in the extracellular area is to mimic secretion and the internalization of Pgk1 and study its potential non-canonical functions. However, this rationale is not clear enough in the introduction section of the manuscript. The authors should justify clearly why the exogenous administration of Pgk1 is relevant.

Author’s response to Q1:

In response to your suggestion, we revised and added more information in the Introduction section as follows: “Particularly, Lin et al. [38] revealed that ePgk1 triggers the reduced phosphorylation of Cofilin at Ser3 (p-Cofilin-S3), a hallmark of growth cone collapse in neuronal cells, through decreasing the signaling pathway of Rac-GTP/p-Pak1-T423/p-MK2-T334/p-Limk1-S323/p-Cofilin-S3, which, in turn, enhances neurite outgrowth of motor neurons in a manner functionally independent from its intracellular, canonical role as a supplier of energy. This evidence suggests that the level of p-Cofilin-S3 is a biomarker determinative of neuronal cell development. Interestingly, Tseng et al. [39] reported that the level of p-Cofilin is increased in MPP+-treated primary mesencephalic cells, suggesting that it serves as a candidate biomarker of MPP+‐induced neurite length reduction. Therefore, it would be worthwhile to investigate whether addition of ePgk1 enables the reduced expression of p-Cofilin in MPP+/MPTP-treated cells, resulting in alleviating neuronal damage. In this study, we demonstrated that extracellular administration of Pgk1 could also serve as a neuron-protective substance for neurotoxin-treated dopamine neurons in the brain. It would be very interesting to conduct further investigation of this issue since a conclusive finding would be a significant step forward in understanding whether ePgk1 could also protect dopaminergic neurons from the degeneration and cell death that occur in the brain of PD patients.” (Please see lines 101-117)

  1. The colocalization of Pgk1 with TH antibody is not apparent. The Pgk1-Flag seems to be localized in the extracellular space rather than the cytoplasm of DA neurons. The images (Figure 4-A, B, C) are not resolved enough to confirm the colocalization. The signal from dat: EGFP is overexposed. The marked areas, especially the dotted circle, lack any specific signal! The authors need to provide better-quality images to support their data.

Author’s response to Q2:

Based on the evidence shown in the attached Supplementary Figure 2 in this study, we believe that a receptor specific for ePgk1 is located at the cell membrane of NSC34 neuronal cells. In fact, we have already revealed that the interaction between ligand ePgk1 and such receptor could trigger reduction of the signaling pathway p-Pak1-T423/p-MK2-T334/p-Limk1-S323/p-Cofilin-S3, in turn enhancing neurite outgrowth (Fu et al., unpublished data). Therefore, the colocalization of Pgk1 with TH antibody would not be apparent since ePgk1-Flag localizes to either the extracellular space or cell membrane, rather than the cytoplasm, of DA neurons, as shown in Figure 4.

As requested, we have improved the quality of images in Figure 4 by adjusting the image capture and removing the dotted circle to focus on the neurite position, thereby avoiding interference of the dopamine neuron cell body signal. However, we apologize for incorrectly describing “colocalization” in the Abstract and legend of Figure 4 which led to the misunderstanding. In the revised manuscript, we changed “colocalization” to “location.” (Please see line 19 and 287)

  1. The authors have also not provided evidence that ePkg1 is readily absorbed into the SH-SY5Y cells without any protein penetration strategies.

Author’s response to Q3:

Similar to my response to Q2 above, let me clarify. Since ePgk1-Flagwas localized either in the extracellular space or cell membrane, rather than the cytoplasm, of DA neuron, we could not provide any evidence that ePgk1 was internalized into the cells through protein penetration strategy.

  1. The authors have used Students-T-tests in their statistical analysis. The Student t-test is used to compare the means between two groups, whereas ANOVA should be used to compare the means among three or more groups. The materials and methods section lacks a subsection on statistics. The authors should describe the parameters used to employ individual statistical tests and explain this in the materials and methods section.

Author’s response to Q4:

Thank you for pointing out this mistake. We added ANOVA statistical analysis to compare the means among three groups shown in Figure 1B. We also described the parameters used in the statistical tests in the Materials and Methods section. (Please see lines 618-625)

  1. The authors have not referenced the statement (Sentence 261) that DA neuronal loss leads to poor swimming ability. Dopamine acts as an excitatory and inhibitory neurotransmitter with contradicting effects on motility.

Author’s response to Q5:

Thank you for pointing out this mistake. We have now cited Kalyn et al. [26] (https://doi.org/10.3390/biomedicines8010001) to support that DA neuronal loss leads to poor swimming ability.

Minor corrections

  1. The result section needs to be checked for grammatical errors.

Figure 7 A: The alpha-tubulin band does not correspond to the Bcl-2 band in figure 1. Author’s response to 1:

  • We carefully checked for grammatical errors in the Results section.
  • In response to your comment, we replaced Figure 7 A with a new one in which the alpha-tubulin band corresponds to the Bcl-2 band in the panel 1 of this figure.

  1. Sentence 18: Time, not times.

Author’s response to 2: This word was corrected. Thank you.

  1. Sentence 31: cell death of dopaminergic neurons- death of dopaminergic neurons or dopaminergic neuronal death.

Author’s response to 3: This has been changed to death of dopaminergic neurons. Thank you.

  1. Sentence 31 substantia nigra pars compacta in italics.

Author’s response to 4: Corrected it. Thank you.

  1. Sentence 78: PD-like Zebrafish model not PD-like zebrafish.

Author’s response to 5: Thank you. We corrected to PD-like Zebrafish model.

  1. ROS expression: Reactive oxygen species levels, not expression. Expression is associated with gene or protein levels.

Author’s response to 6: Thank you. We corrected to reactive oxygen species (ROS) levels.

  1. MPP+ or MPTP treatment in SH-SY5Y cells?

Author’s response to 7: MPP+

Round 2

Reviewer 1 Report

In their revised version, the authors have markedly improved the quality of their manuscript. My previous comments were fairly well addressed although I still do not agree 100% of the time with the authors. In the end, this is an interesting study and I support its publication.

The English still needs some minor work but perhaps can this be addressed by the journal.

Reviewer 2 Report

The authors have addressed the comments satisfactorily.